# Enhanced THz Circular-Polarization Detection in Miniaturized Chips with Chiral Antennas

Fangzhe Li [1,2,3], Jing Zhou [1,*], Jie Deng [1], Jinyong Shen [1,2], Tianyun Zhu [1,2], Wenji Jing [1,2], Xu Dai [1,2], Jiexian Ye [1], Yujie Zhang [1,2], Junwei Huang [1,2] and Xiaoshuang Chen [1,3,*]

[1] State Key Laboratory of Infrared Physics, Shanghai Institute of Technical Physics, Chinese Academy of Sciences, Shanghai 200083, China; jingwenji@mail.ustc.edu.cn (W.J.); zhangyujie232@mails.ucas.edu.cn (Y.Z.)
[2] University of Chinese Academy of Sciences, Beijing 100049, China
[3] School of Physical Science and Technology, ShanghaiTech University, Shanghai 201210, China
[*] Correspondence: jzhou@mail.sitp.ac.cn (J.Z.); xschen@mail.sitp.ac.cn (X.C.)

**Abstract:** Recent advancements in terahertz (THz) wave technology have highlighted the criticality of circular-polarization detection, fostering the development of more compact, efficient on-chip THz circular-polarization detectors. In response to this technological imperative, we presented a chiral-antenna-integrated GaAs/AlGaAs quantum well (QW) THz detector. The chiral antenna selectively couples the incident light of a specific circular-polarization state into a surface-plasmon polariton wave that enhances the absorptance of the QWs by a factor of 12 relative to a standard 45° faceted device, and reflects a significant amount of the incident light of the orthogonal circular-polarization state. The circular-polarization selectivity is further enhanced by the QWs with a strong intrinsic anisotropy, resulting in a circular-polarization extinction ratio (CPER) as high as 26 at 6.52 THz. In addition, the operation band of the device can be adjusted by tuning the structural parameters of the chiral structure. Moreover, the device preserves a high performance for oblique incidence within a range of ±5°, and the device architecture is compatible with a focal plane array. This report communicates a promising approach for the development of miniaturized on-chip THz circular-polarization detectors.

**Keywords:** terahertz; circular-polarization detection; miniaturized on-chip detectors; quantum well; chiral antenna; surface plasmon polariton; circular-polarization extinction ratio





## 1. Introduction

Circularly polarized light (CPL) is characterized by its high polarization fidelity during transmission [1], unique interaction with chiral substances [2], and superior ability to carry quantum information through photon spins [3]. Thus, CPL has found many important applications in a lot of fields including optical communication [4], astronomical imaging [5], plasmonic sensing [6], and quantum information processing [7–9]. In all these applications, circular-polarization detection is always a critical technique. The miniaturization trend in optoelectronics has given rise to compact integrated circularly polarized detectors. These devices are intended to replace traditional setups that involve discrete polarizers and wave plates. This movement has been greatly propelled by advances in plasmonics, especially through the study of plasmon polaritons—electromagnetic waves that interact with free electron oscillations on metal surfaces. The realm of plasmonics has expanded considerably, pioneering novel approaches for manipulating light at the nanoscale [10–14]. Plasmonic structures have become increasingly prevalent in the manipulation of light coupling for photoelectric detectors, as shown in numerous studies [15]. These structures have the remarkable ability to convert incident light into surface plasmon polariton (SPP) waves, resulting in an intensified local field confined to subwavelength or even deeper subwavelength regions adjacent to metal/dielectric interfaces. When seamlessly integrated

with infrared detectors, plasmonic structures offer the promise of enhanced and controllable light absorption within the detection materials proximal to these structures. In contrast to dielectric photonics structures, plasmonic structures exhibit a distinct advantage in their integration with infrared detectors, as they can simultaneously function as detector contacts while mediating light coupling. Despite these promising attributes, substantial challenges persist in realizing the full potential of this technological pathway. Chief among these is the limited coupling efficiency, which currently hinders widespread practical application. With the continuous development of research in the field of surface plasmons, their applicability to the field of photoelectric detection has also begun to increase. Among the potential applications, the combination of metal surface plasmon and quantum-well photoelectric detection produces many novel bright spots. With the progress in micro-nano fabrication technologies, chiral plasmonic antennas have been proposed to integrate with detection materials to form on-chip circular-polarization detectors [15,16]. These antennas are engineered to selectively couple left-handed circularly polarized (LCP) or right-handed circularly polarized (RCP) light into enhanced local fields and concurrently reflect the CPL with the opposite handedness [17–23]. While substantial research has been concentrated on integrated circular-polarization detectors in the visible-to-mid-infrared range, the THz band remains relatively underexplored. The THz band, noted for its low photon energy [24,25], robust penetration capabilities [26,27], expansive communication bandwidth [28,29], and superior spatial and temporal resolution [30–32] compared to microwaves, holds significant potential for advancing imaging, spectroscopy, and wireless communication [33]. Thus, the development of miniaturized THz circular-polarization detectors, which addresses the challenges posed by bulky non-integrated detectors and the scarcity of suitable polarization optics in the THz range, is not only of great significance but also imminent.

This report presents a revolutionary miniaturized THz circular-polarization detector, consisting of multiple GaAs/AlGaAs quantum wells (QWs) sandwiched between a chiral plasmonic antenna array and a metal plane. The device selectively couples the incident LCP or RCP light into a substantially enhanced surface plasmon polariton (SPP) wave and reflects the CPL with the opposite handedness to a significant degree. The intensified local field at the QWs enhances the absorption of the CPL with the selected handedness while the reflection of the CPL with the opposite handedness reduces the absorption of this type of light in the QWs, thereby efficiently distinguishing between the two circular-polarization states in the photocurrent of this device. The GaAs/AlGaAs QWs only absorb light with an electric field component orthogonal to the multiple layers, thus improving the circular-polarization extinction ratio (CPER) [34]. This dual-polarization selection effect results in an impressive CPER of 26 at 6.52 THz ($\lambda$ = 46 µm), surpassing the performance of existing integrated THz circular-polarization detectors [35]. In addition, the SPP wave at the QWs yields a 12-fold increase in the absorption of the CPL with the selected handedness, compared to a reference 45°-edge-facet-coupled device. By meticulously adjusting the structural parameters of the device, the absorption peak is tunable between 6.42 and 6.55 THz, accompanied by improved absorption and polarization selectivity across various incident angles. This innovative design represents a significant leap forward in the development of high-performance, compact on-chip THz circular-polarization detectors.

## 2. Device Structure

The GaAs/AlGaAs Quantum Well (QW) detector, central to infrared research, fuses semiconductor physics with quantum mechanics. This advanced device consists of a thin GaAs layer enclosed by two AlGaAs layers, forming a quantum well that spawns subbands. Its operation relies on aligning the energy gap with incident photons, enabling photoconductivity and boosting conductivity through inter-subband transitions. The combination of GaAs and AlGaAs, both III-V semiconductors, is crucial. The heterostructure, created by the adjustable bandgap of AlGaAs, enforces quantum confinement, limiting electron and hole movement in the vertical direction but allowing lateral motion. This results in discrete energy levels, so-called subbands, in the quantum well [36]. A distinctive feature of the

on-chip THz circular-polarization detector is the composite structure of the chiral antenna and the III-V semiconductor layers (Figure 1a). The semiconductor layers include 27 stacks of $Al_{0.04}Ga_{0.96}As$ (75 nm)/GaAs (15 nm) quantum wells with an additional $Al_{0.04}Ga_{0.96}As$ (75 nm) barrier layer, a 0.5 μm-thick top GaAs contact layer and a 2.43 μm-thick bottom one (Figure 1b). The semiconductor layers are epitaxially grown and then sandwiched by an Au reflective layer and a two-dimensional (2D) chiral metamaterial layer with the help of a substrate removal technique [34,37–40]. As illustrated in Figure 1c, the 2D chiral metamaterial is defined by $P_x$ = 13 μm, $P_y$ = 14.91 μm, $W$ = 3.14 μm, and $L$ = 8.61 μm. $P_x$ and $P_y$ are the periods in the *x*- and *y*-directions, respectively. In each period of the 2D chiral metamaterial, the center of the twisted Au strip must coincide with the center of the rectangle. The 2D chiral structure is composed of two neighboring, but not overlapping metal ribbons. Then, the secondary parameters ($\theta$ and $L_c$) can be determined using $P_x$, $P_y$, $W$, and $L$. Figure 1d shows a three-dimensional schematic illustration of one period of the chiral metamaterial integrated QWs under the oblique incidence of a THz wave.

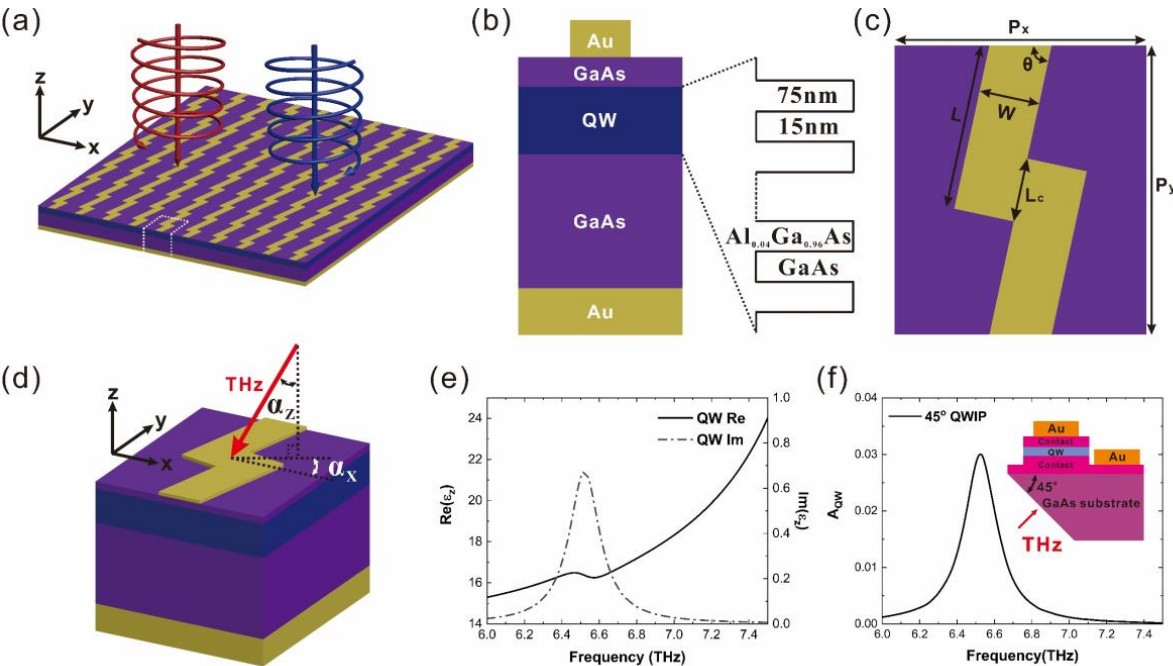

**Figure 1.** (**a**) Three-dimensional schematic of the device. (**b**) Cross-section along the *z*-axis from unit of (**a**). (**c**) Top view of the device unit. (**d**) Illustration of tilt-angle concept. (**e**) Spectra of real and imaginary parts of the relative permittivity along *z*-direction ($\varepsilon_z$) of the QW material. (**f**) Absorption spectrum of a 45°-edge facet and device structure. The 45°-edge-facet-coupled device features a QW material substrate, angled at 45 degrees, with etched and evaporated upper and lower electrode layers accessible for wire bonding. When incident light strikes the ground surface, the *z*-direction electric field component induces QW light absorption.

The QWs are designed to detect THz light with a subband gap corresponding to the photon energy of 46 μm. Due to the selection rule of the inter-subband transition, the QWs can only absorb light possessing a z-component electric field. Thus, the QWs can be regarded as a uniaxial effective medium, described by a permittivity tensor $\varepsilon_{QW} = diag(\varepsilon_x, \varepsilon_y, \varepsilon_z)$, where $\varepsilon_x$ and $\varepsilon_y$ are the same as the permittivity of GaAs ($\varepsilon_{GaAs}$), and $\varepsilon_z$ is formulated as a Lorentz oscillator ($\varepsilon_z = \varepsilon_{GaAs} + \frac{\varepsilon_{GaAs}f_{12}\omega_p^2}{\omega_{12}^2-\omega^2-i\omega\gamma}$) [41] that describes the inter-subband transition. Key parameters in this formulation include the oscillator strength ($f_{12}$), the 2D effective plasma frequency ($\omega_p$), the optical transition frequency ($\omega_{12}$), and the relaxation frequency ($\gamma$) [42]. The frequency dependent Re ($\varepsilon_z$) and Im ($\varepsilon_z$) are presented in Figure 1e. The peak in the frequency dependent Im ($\varepsilon_z$) curve corresponds to the energy of the inter-

subband transition and thus the energy of the detected photons. Here, the peak frequency is 6.52 THz, indicating that the QWs are designed for detecting light with a frequency of around 6.52 THz. For the incident light polarized within the *x-y* plane, the QWs behave similarly to GaAs, which is a transparent dielectric in the long-wave infrared range. For the incident light polarized in the *z*-direction, photocarriers can be excited in the QWs due to the inter-subband transition. Since QWs are insensitive to normally incident light, they are typically made into a 45°-edge-facet-coupled device for photoresponse characterization [43]. As shown in Figure 1f, the peak absorptance of the QWs in the 45°-edge-facet-coupled device is about 0.03. Additionally, the dielectric constant of Au is modeled using the Drude model [44].

## 3. Result and Analysis

Figure 2a,b demonstrate that in the frequency range from 6.37 to 6.68 THz, the absorptance of LCP light by the QWs significantly exceeds that of RCP light. Notably, at 6.52 THz ($\lambda$ = 46 µm), the CPER attains a maximum of 26, coinciding with the peak absorptance of 0.373 for LCP light. Figure 2c illustrates how LCP light resonantly excites a pronounced local field between the chiral antenna and the metal plane. This local field, characterized by a substantial $E_z$ component, overlaps the QWs in the photosensitive region, resulting in a marked increase in absorptance. The genesis of this local field is an SPP wave at the interface between the chiral antenna and the GaAs top contact layer. For normally incident light, two SPP waves propagating in the positive *y*-direction and negative *y*-direction are excited. Conversely, Figure 2d reveals that RCP light fails to effectively excite the SPP wave, resulting in a considerably lower field intensity at the QWs and diminished absorptance. This remarkable discrepancy in the absorption rates of LCP and RCP light facilitates high circular-polarization discrimination. Furthermore, the absorptance of LCP light is observed to be over 12 times greater than that of a conventional 45°-edge-facet-coupled device. Therefore, the device under discussion exhibits not only a high capacity for circular-polarization discrimination but also an amplified absorption in the case of the principal circular polarization.

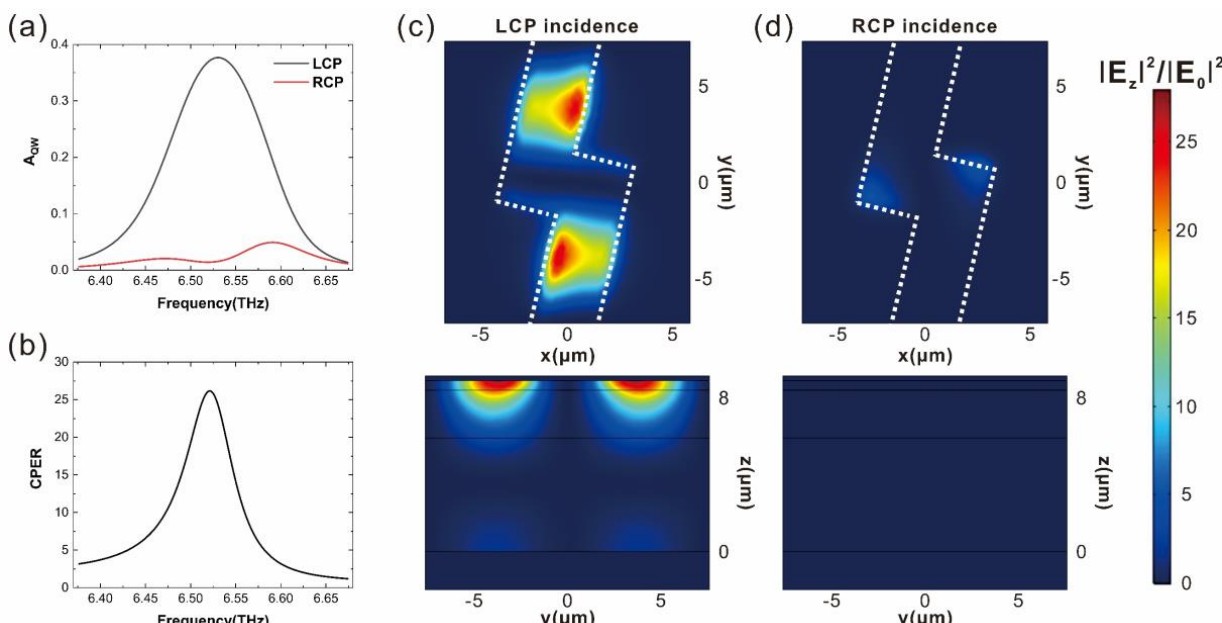

**Figure 2.** (**a**) the absorption spectrum within the QW layer is depicted under the influence of LCP and RCP light. (**b**) is dedicated to presenting the CPER spectrum. (**c,d**) focus on the spatial distribution of the electromagnetic field. (**c**) illustrates the field distribution in the x-y plane at the center of the QW layer, while (**d**) visualizes the field distribution in the y-z plane along the central axis of the structural unit. (**c,d**) are evaluated under the distinct conditions of LCP and RCP incidence.

In the context of assessing detection performance under low-temperature conditions, the noise current can be approximated by the formula $i_{noise} = \sqrt{4i_{dark}eg\Delta f}$ [45]. The detectivity is calculated using the equation $D^* = R\sqrt{A}/i_{noise}$, where $A$ refers to the photosensitive area and $R$ indicates the responsivity. Based on a previous study [46], the performance of our proposed detector is estimated. For the selected quantum well material, the responsivity of the 45°-edge-facet-coupled device is noted as 0.049 A/W. The integration of the specifically designed antenna and a bottom Au reflector has been observed to increase the responsivity up to 0.61 A/W. With a typical dark current of $4.06 \times 10^{-10}$ A, the noise current is calculated to be approximately $9.82 \times 10^{-15}$ A $Hz^{-1/2}$. Consequently, the responsivity of the detector could reach $8.68 \times 10^{10}$ cm $Hz^{1/2}$ $W^{-1}$.

The SPP is effectively excited when the incident light acquires an additional wave vector $(2\pi/P_y)$ from the grating structure, aligning with the propagation constant of the SPP wave, $\beta_{spp} = (2n\pi/\lambda_0)\sqrt{\varepsilon_m\varepsilon_d/(\varepsilon_m + \varepsilon_d)}$, where $n = \pm1, 2, 3\ldots$ Here, $\lambda_0$ represents the free-space wavelength of the incident light, $\varepsilon_m$ the relative permittivity of the metallic material, and $\varepsilon_d$ the relative permittivity of the dielectric medium. The disparity in the excitation efficiency of the SPP wave between LCP and RCP light is ascribed to the interference between the co-polarized and cross-polarized radiation of the twisted metal strip. The circular-polarization selectivity in asymmetric metamaterials is discernible by examining the interference of co-polarized and cross-polarized components in the reflection field [33]. CPL is decomposable into two orthogonal electric fields, $E_x$ and $E_y$, with a phase difference of $\pi/2$. Upon the incidence of a linearly polarized light field, either $E_x$ or $E_y$, on asymmetric metamaterials, the resultant reflection field comprises both co-polarized fields ($r_{xx}E_x$ or $r_{yy}E_y$) and cross-polarized fields ($r_{yx}E_x$ or $r_{xy}E_y$). The coefficients $r_{xx}$ and $r_{yy}$ correspond to the reflection coefficients for co-polarization, whereas $r_{yx}$ and $r_{xy}$ correspond to those for cross-polarization. This reflection process is quantifiable via the Jones matrix [34,47].

$$\begin{pmatrix} E_x^r \\ E_y^r \end{pmatrix} = \begin{pmatrix} r_{xx} & r_{xy} \\ r_{yx} & r_{yy} \end{pmatrix} \begin{pmatrix} E_x \\ E_y \end{pmatrix} \tag{1}$$

Figure 3a,b illustrate the amplitude and phase of calculated reflection coefficients, respectively. The investigation into the interference between co-polarized and cross-polarized reflected fields for LCP and RCP light was conducted using vector field diagrams, as shown in Figure 3c–f. At the resonant frequency of the SPP wave, the co-polarized reflected field ($r_{xx}E_x$ or $r_{yy}E_y$) exhibits an out-of-phase relationship with the cross-polarized reflected field ($r_{yx}E_x$ or $r_{xy}E_y$) for LCP light, as depicted in Figure 3c,d. This phase discrepancy results in a diminished total reflection ($E_{x,sum}$ or $E_{y,sum}$), implying that a greater portion of the incident power is transferred into the SPP wave, thereby amplifying the local field at the QWs. In contrast, for RCP light, the co-polarized field ($r_{xx}E_x$ or $r_{yy}E_y$) aligns in phase with the cross-polarized field ($r_{yx}E_x$ or $r_{xy}E_y$), as shown in Figure 3e,f. This alignment leads to an increased total reflection ($E_{x,sum}$ or $E_{y,sum}$), indicating that the incident light is less efficient in exciting the SPP wave, which in turn results in a weaker local field at the QWs [34,48]. In addition, the combined structure of the bottom metal plane and chiral antennas forms a cavity, significantly enhancing the contrast between LCP and RCP light in a similar way to film interference [34,48].

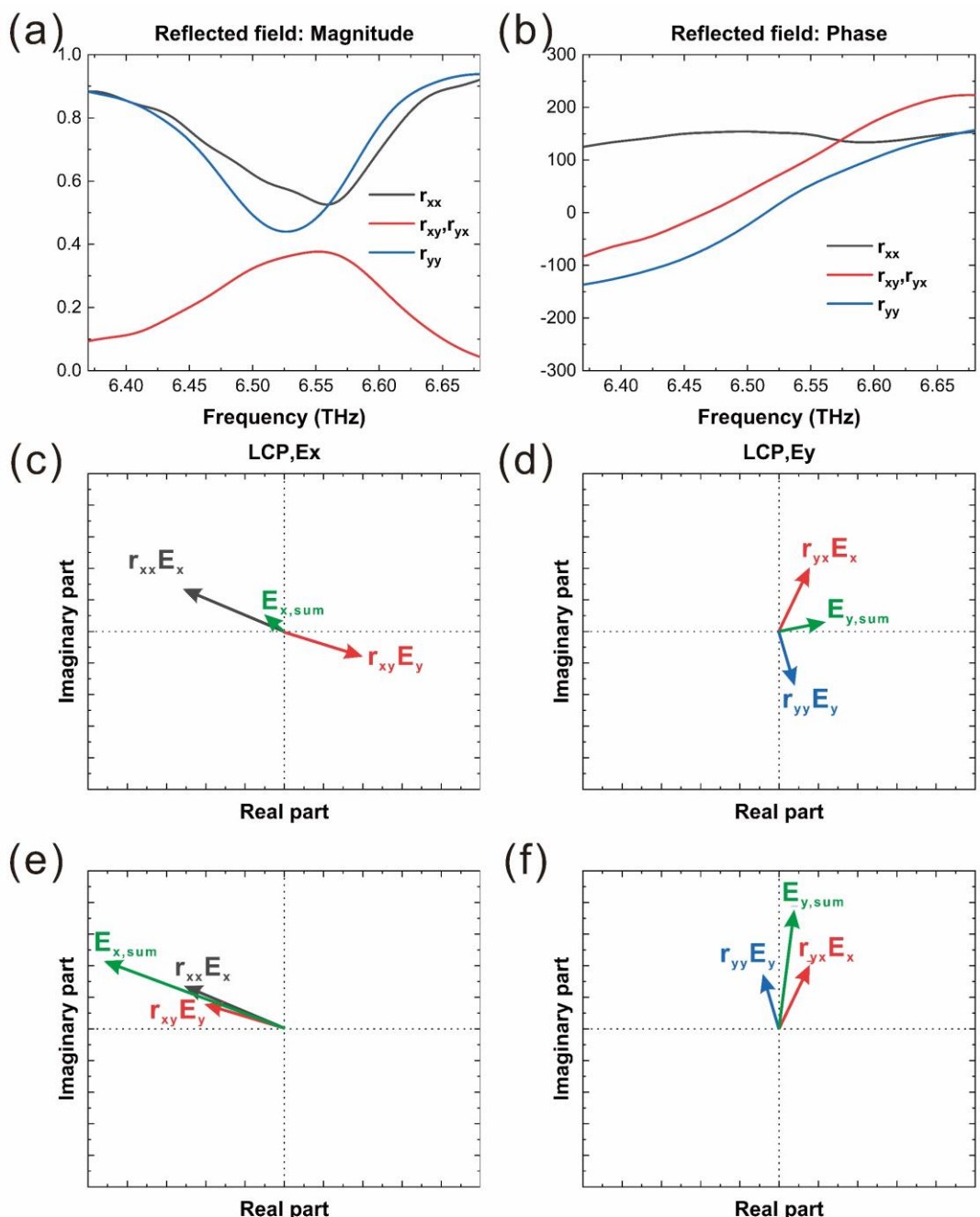

**Figure 3.** (**a**,**b**) illustrate the amplitude and phase of reflection coefficients. (**c**–**f**) show vector plots of interference: destructive in (**c**,**d**) and constructive in (**e**,**f**), between the unconverted ($r_{xx}E_x$ or $r_{yy}E_y$) and converted reflected fields ($r_{xy}E_y$ or $r_{yx}E_x$) at 6.52 THz resonance.

## 4. Discussion

By modifying the physical parameters of the engineered chiral structure, it is possible to tune the absorption peak within the QWs. The adjustment of the resonant frequency to excite the SPP wave can be achieved by altering the parameters $P_x$ and $P_y$. As shown in Figure 4, increasing $P_y$ leads to a longer wavelength for the absorption peak of the LCP light. In order to fulfill the momentum-matching condition $\beta_{spp} = (2n\pi/\lambda_0)\sqrt{\varepsilon_{eff}\varepsilon_d/\left(\varepsilon_{eff}+\varepsilon_d\right)} = 2\pi/P_y$, where $n = \pm1, 2, 3\ldots$, a larger $P_y$ corresponds to a longer free-space wavelength ($\lambda_0$) [49]. The influence of $P_x$ is analyzed by considering the 2D chiral metamaterial layer as an effective medium as a mixture of Au and air [50].

Keeping $P_y$ fixed and increasing $P_x$ makes the effective medium less metallic and thus the effective permittivity ($\varepsilon_{eff} = f\varepsilon_{Au} + (1-f)\varepsilon_{air}$) less negative, resulting in a red shift in the absorption peak. $f$ denotes the Au fraction in the effective medium. In summary, increasing either $P_y$ or $P_x$ leads to a redshift trend in the absorption peaks of the designed structures.

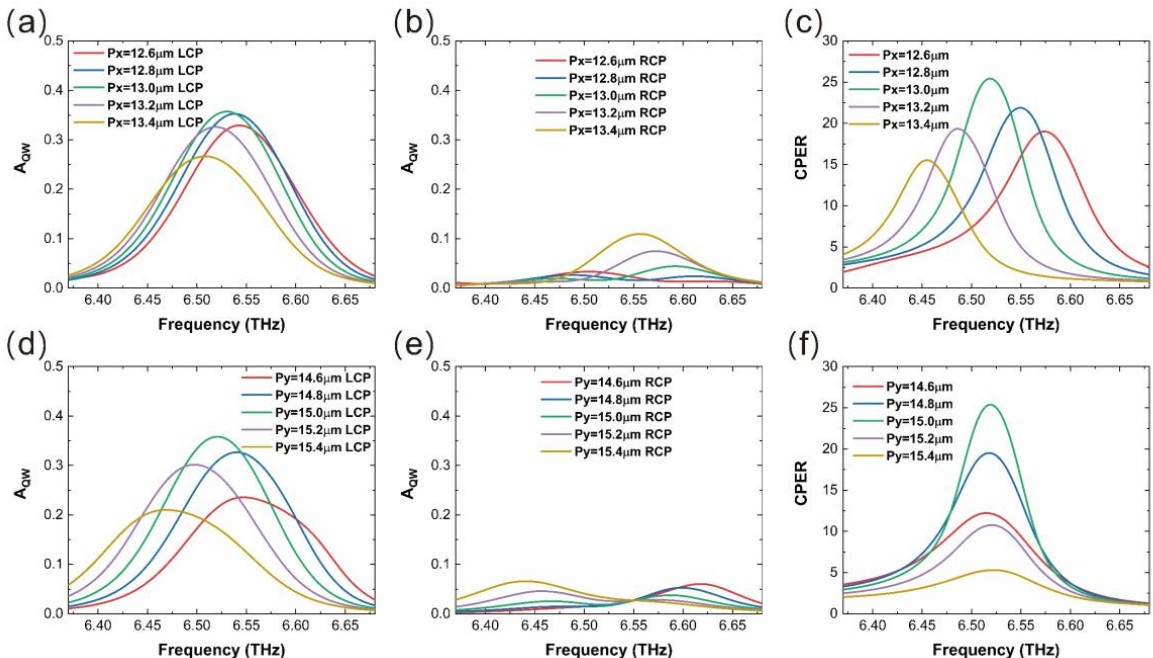

**Figure 4.** The absorption and CPER spectra, derived from modifications in the structural period along two axes. (**a**–**c**) illustrate the outcomes of altering the period $P_x$ in the x-direction, and (**d**–**f**) display the effects of adjusting the period $P_y$ in the y-direction.

Integrating twisted metal strips of different structural parameters together is a way to increase the bandwidth of the circular-polarization detector. We conducted additional research and analysis on the effects of $P_x$ and $P_y$ on the circular-polarization-selective absorption of the device. Exploiting the impact of periodicity variations, it is expected that arranging twisted metal strips with different $P_x$ and $P_y$ values on the same mesa would facilitate the broadening of the bandwidth of circular-polarization-selective absorption. Figure 5 presents the spectra for the LCP absorption, RCP absorption, and CPER for five combinations of $P_x$ and $P_y$. Basically, as $P_x$ and $P_y$ increase, the absorptance peak for the principle circular-polarization light (LCP in this case) and the peak of CPER are both redshifted. Therefore, if we integrated these five twisted metal strips together, the bandwidth of circular-polarization-selective absorption would be expanded from 1.35 THz to 2.25 THz. This strategy promotes the bandwidth expansion of the device.

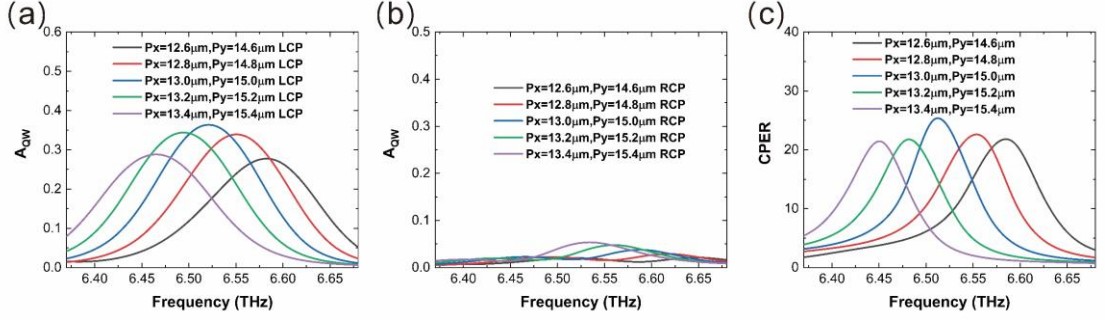

**Figure 5.** Absorption spectra for LCP (**a**) and RCP (**b**) incident and corresponding CPER (**c**) spectra derived from the variation of the structural period along both axes simultaneously.

The research further encompasses an analysis of how *L* and *W* influence the circular-polarization-dependent absorption. Figure 6a,b elucidate the effects of varying *L* on the absorption spectra for LCP and RCP light, as well as on the CPER. As shown in Figure 6c, when *L* is approaching 7.8 μm, the CPER correspondingly decreases to almost 1. The co-polarization and cross-polarization in the reflected wave caused by the structure cannot produce coherent cancellation or coherent construction here. A similar trend is observed with variations in *W*, as shown in Figure 6d–f. The decrease in *W* results in the obvious decrease in structural chirality. However, despite any changes in *W* or *L*, an interfacial overlapping region, the part defined by $L_c$ in Figure 1c, persists between the two ribbons forming the structure, thereby making it impossible to reach the point where chirality thoroughly disappears. As *W* increases, the structure's chirality of structure design intensifies, leading to a corresponding rise in the CPER. Notably, the CPER attains its peak value when *W* = 3 μm. In the realm of cross-polarization radiation, it is observed that the parameters *L* and *W* play pivotal roles in determining the CPER. These circular-polarized-dependent optical properties arise from the co-polarized and cross-polarized oscillatory currents in the 2D chiral metamaterial, functioning akin to intersecting dipole antennas. The structural parameters *L* and *W* determine the amplitudes and phases of these dipole antennas, significantly influencing the CPER. A critical observation is made at specific parameters, precisely *L* = 8.61 μm and *W* = 3.14 μm, where the most significant variation in the field strength is noted, as illustrated in Figure 2c,d. For LCP light, the cross-polarization radiation engages in destructive interference with the principal polarization radiation. In contrast, for RCP light, a constructive interference is noted with the principal polarization radiation, culminating in an optimized maximum CPER.

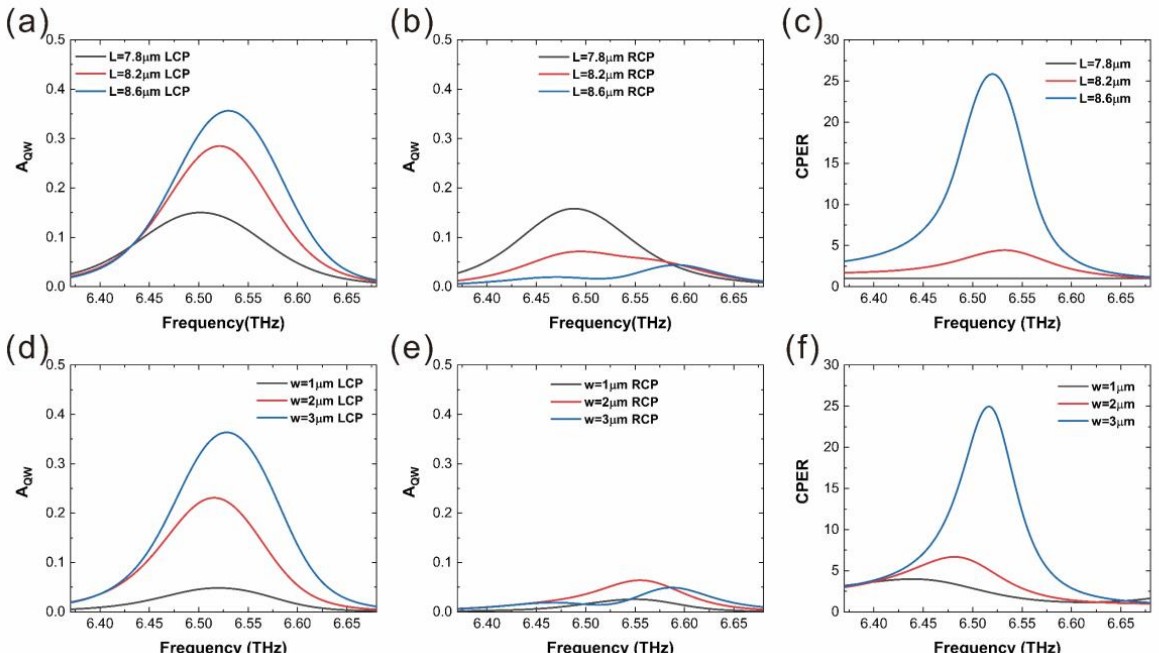

**Figure 6.** Absorption and CPER spectra for LCP and RCP light, categorized by different parameter dependencies. (**a**–**c**) correspond to the *L*-dependent spectra, (**d**–**f**) illustrate the *W*-dependent spectra.

The incident angle dependence of the chiral antenna integrated on-chip THz circular-polarization detector was also studied. During oblique incidence, the incident light acquires a lateral wave vector, equating to lateral momentum. Consequently, the momentum-matching condition is modified to $2n\pi/P_y + k_y = \beta_{spp}$, where $n = \pm 1, 2, 3 \ldots$ For the incident light inclining in the *y-z* plane within a 0°~10° angle range, the lateral wave vector $k_y$ created by the oblique incidence, which is $2.37 \times 10^4$ 1/m, is much smaller than that acquired from the grating of the chiral structure, which is $4.21 \times 10^5$ 1/m. Consequently, the peak absorption in the SPP mode remains stable despite changes in the incident angle.

In the *x-z* plane, alterations in the incident angle do not affect the resonance wavelength of the SPP mode, leading to only minor variations in the absorption peak position across different incident angles. However, the incident angle affects both principal and cross polarization interference, causing the CPER to decrease as the angle increases. As shown in Figure 7, the device maintains remarkable performance consistency within an oblique incidence angle. The CPER remains higher than 10 when the incident angle varies ±5°. This incident angle dependence ensures that our device is useful in practical applications.

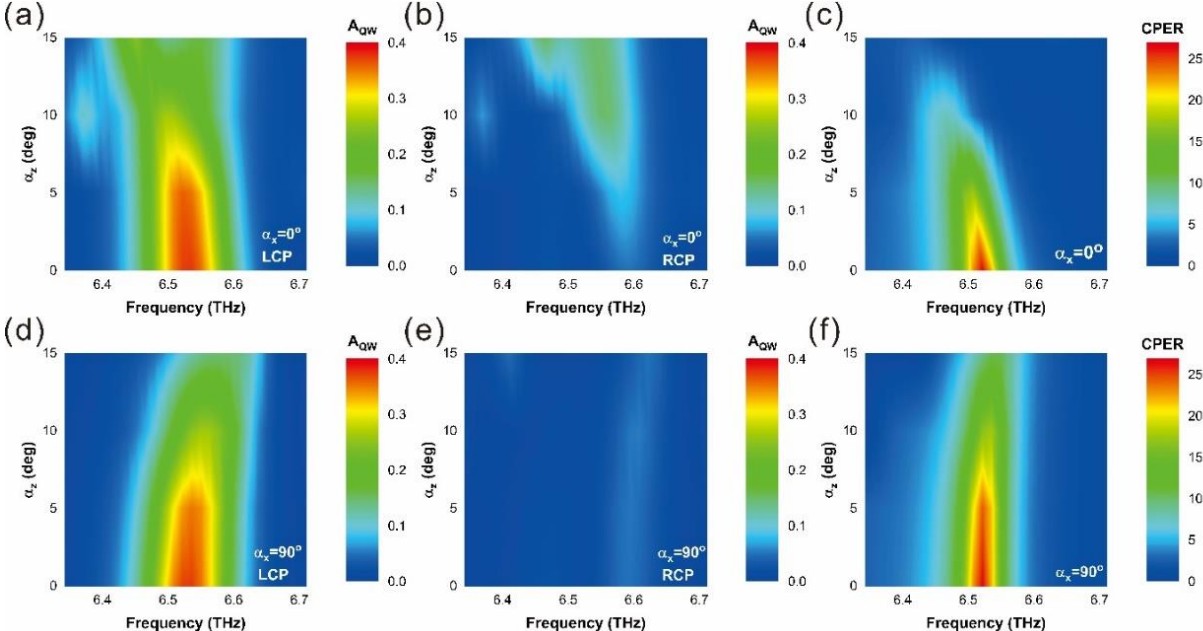

**Figure 7.** Absorption and CPER spectra of the structure under oblique incidence. (**a**–**c**) detail the spectra with an incident angle inclined towards the *x*-axis, and (**d**–**f**) display the spectra for an angle inclined towards the *y*-axis.

Polarization-sensitive materials provide a straightforward and efficient means of realizing polarization detection. These materials, including anisotropic materials, offer a compact detector design that reduces the need for extra fabrication steps compared to that of traditional detectors. Recent studies have highlighted the potential of a range of anisotropic and chiral materials in polarization detection, including van der Waals materials and heterostructures for linear polarization, as well as chiral hybrid perovskites, topological materials, and other circular-polarization-sensitive materials that can distinguish the handedness of circularly polarized light. Recent developments in micro- and nanofabrication techniques have opened up new possibilities, enabling the integration of polarization-selective optical coupling structures, such as an asymmetric plasmon optical-coupling structure and chiral-structure-integrated dielectric optical waveguide, with polarization-sensitive materials to significantly enhance detector performance. This integration, combined with the anisotropic absorption properties of materials, has led to a marked improvement in polarization discrimination [51]. Integrating with different micro-nano optical coupling structures can lead to significant progress in miniaturized on-chip integrated photodetection. However, research into the structures mentioned above has typically been focused on the visible or near-infrared and mid-infrared spectrum [16,34,52–61] and occasionally extends into the long-wave infrared region. In this study, an asymmetric plasmon optical-coupling structure was used and integrated with anisotropic-quantum-well materials to achieve high-performance circular-polarization detection. The primary highlight of this study lies in the device structure designed for superior circular-polarization detection performance within the THz band. This is achieved through the dual polarization selection effect of the chiral structure and the QWs with significant anisotropy, the response of the

QW subband transition to THz wave, and the selective coupling of the plasmonic chiral antenna to circularly polarized light.

## 5. Conclusions

In summary, in the realm of THz technology, a groundbreaking advancement is presented through the development of an enhanced THz circular-polarization detector using on-chip integrated chiral antennas. This innovation capitalizes on a composite structure, central to which is a QW infrared detection material, strategically situated between a chiral plasmonic antenna array and a metallic plane. The device efficiently couples the incident circularly polarized waves of a specific handedness into a cavity-enhanced SPP wave. This coupling enhances the absorptance of the QWs by 12 times compared with that of the QWs in a 45°-edge-facet-coupled device. In contrast, circularly polarized waves of opposing handedness are reflected at a high rate, leading to a decrease in light absorption in the QWs, thus enabling effective discrimination between different circular-polarization states. A notable aspect of this detector is the employment of GaAs/AlGaAs QWs, which are responsive only to light with an electric field component perpendicular to the QWs, providing an additional polarization selection and thus resulting in an impressive CPER, reaching up to 26. In addition, the peak absorption of the principle circularly polarized light and the peak CPER can be adjusted by tuning the structural parameters of the 2D chiral structure. Further, the device preserves a high performance for oblique incidence within a range of $\pm 5°$. Such attributes of the design highlight its significant potential to contribute to the evolution of efficient, miniaturized on-chip THz circular-polarization detectors, marking a notable stride in THz technological advancements. This advancement will undoubtedly enhance the application and progression of terahertz detection technology across a broad spectrum of fields. These include, but are not limited to, wireless and optical communication, security screening, and biomedical imaging, as well as chemical and biomolecular detection.

**Methods.** For the evaluation of the newly proposed structure, its performance was assessed using the three-dimensional finite difference time domain (FDTD) method, as delineated in Lumerical Inc. [62] and COMSOL Multiphysics [63]. This assessment entailed the simulation of a unit cell, for which periodic boundary conditions were implemented along the x and y axes, complemented by perfectly matched layers (PML) along the z axis. The simulation encompassed a model unit with a total area measuring $13 \times 14.91 \ \mu m^2$. The structure in question encompasses three layers, each serving a distinct function. The uppermost layer, pivotal for chiral light absorption, possesses a 50 nm thickness. In stark contrast, the structure's base features a metal reflector, 2 $\mu m$ thick. Interposed between these two is an absorption layer, a complex arrangement consisting of an upper electrode layer (0.5 $\mu m$ thick), a 27-period quantum well layer (2.505 $\mu m$ thick), and a lower electrode layer (5.945 $\mu m$ thick). The quantum-well layer itself comprises 27 15-nm GaAs potential wells interspersed with 28 75 nm $Al_{0.04}Ga_{0.96}As$ barriers. The electrodes, both upper and lower, play a crucial role in facilitating the application of an external electric field to the quantum well layer, thereby enabling control over its optical characteristics.

**Author Contributions:** In this collaborative effort, the concept was jointly developed by F.L., J.Z. and X.C. The numerical simulations were executed by F.L., while the data analysis was the collective effort of F.L. and J.Z., who also took the lead in manuscript composition. Valuable analytical insights and suggestions were provided by J.D., J.S., T.Z., W.J., X.D., J.Y., Y.Z. and J.H., enhancing the depth and quality of the research. The design of the device structure was further refined with the assistance of J.S. and T.Z. A comprehensive discussion of the project was facilitated by the active participation of all authors, ensuring a holistic approach to the research. The manuscript was primarily written by F.L., with significant contributions from each member of the team, embodying a collaborative spirit. The oversight and guidance of J.Z. and X.C. were pivotal in steering the project to fruition, culminating in a well-coordinated and multi-faceted research endeavor. All authors have read and agreed to the published version of the manuscript.

**Funding:** Strategic Priority Research Program (B) of the Chinese Academy of Sciences (XDB0580000); National Key Research and Development Program of China (2022YFA1404602); National Natural Science Foundation of China (U23B2045, 61975223, 61991442, 62305362); Program of Shanghai Academic Research Leader (22XD1424400); Shanghai Municipal Science and Technology Major Project (2019SHZDZX01); SITP Innovation Foundation (CX-461, CX-522).

**Institutional Review Board Statement:** Not applicable.

**Informed Consent Statement:** Not applicable.

**Data Availability Statement:** The data related to the paper are available from the corresponding authors upon reasonable request.

**Acknowledgments:** The authors extend their gratitude to ShanghaiTech Quantum Device Lab (SQDL) and Nanofabrication facility at the Suzhou Institute of Nano-Tech and Nano-Bionics (CAS) for valuable consultations on the feasibility of the device structure.

**Conflicts of Interest:** The authors declare no conflicts of interest.

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
