# Peer review of "Enhanced THz Circular-Polarization Detection in Miniaturized Chips with Chiral Antennas"

_photonics, doi:10.3390/photonics11020162_

Round 1

Reviewer 1 Report

Comments and Suggestions for Authors

This manuscript presents the THz Circular Polarization Detection method. The paper was well-organized with detailed simulation and measurement results. Other comments:

1. How to further improve the bandwidth of the CP Detector?

2. The authors mentioned that the structure is new. Innovative can be further illustrated.

Reviewer 2 Report

Comments and Suggestions for Authors

Reviewer 3 Report

Comments and Suggestions for Authors

A CP on-chip chiral antennas for THz technology is presented in this article.  The 3-D FDTD method is used for analysis. The proposed detector should be suitable for optical application communications.
